# A model for the intrinsic limit of cancer therapy: Duality of treatment-induced cell death and treatment-induced stemness

Erin Angelini[1], Yue Wang[1,2], Joseph Xu Zhou[3,4], Hong Qian[1], Sui Huang[4]*

**1** Department of Applied Mathematics, University of Washington, Seattle, Washington, United States of America, **2** Institut des Hautes Études Scientifiques, Bures-sur-Yvette, France, **3** Immuno-Oncology Department, Novartis Institutes for BioMedical Research, Cambridge, Massachusetts, United States of America, **4** Institute for Systems Biology, Seattle, Washington, United States of America

* sui.huang@isbscience.org

**Data Availability Statement:** Code used to generate plots in this article and in the Supporting information can be found on GitHub at https://github.com/eeangelini/cancer-model-plots.

## Abstract

Intratumor cellular heterogeneity and non-genetic cell plasticity in tumors pose a recently recognized challenge to cancer treatment. Because of the dispersion of initial cell states within a clonal tumor cell population, a perturbation imparted by a cytocidal drug only kills a fraction of cells. Due to dynamic instability of cellular states the cells not killed are pushed by the treatment into a variety of functional states, including a "stem-like state" that confers resistance to treatment and regenerative capacity. This immanent stress-induced stemness competes against cell death in response to the same perturbation and may explain the near-inevitable recurrence after any treatment. This double-edged-sword mechanism of treatment complements the selection of preexisting resistant cells in explaining post-treatment progression. Unlike selection, the induction of a resistant state has not been systematically analyzed as an immanent cause of relapse. Here, we present a generic elementary model and analytical examination of this intrinsic limitation to therapy. We show how the relative proclivity towards cell death versus transition into a stem-like state, as a function of drug dose, establishes either a window of opportunity for containing tumors or the inevitability of progression following therapy. The model considers measurable cell behaviors independent of specific molecular pathways and provides a new theoretical framework for optimizing therapy dosing and scheduling as cancer treatment paradigms move from "maximal tolerated dose," which may promote therapy induced-stemness, to repeated "minimally effective doses" (as in adaptive therapies), which contain the tumor and avoid therapy-induced progression.

## Author summary

Advance in the war on cancer is concentrated at one single front: more efficient killing of tumor cells, including by targeted or immuno-therapy. However, cells are hard-wired to activate regenerative or protective programs in response to near-lethal stress. Thus, cancer cells not killed during treatment are still stressed and often enter a stem-like state. This

**Funding:** This work was supported by the National Institute of General Medical Sciences (R01GM109964 and R01GM135396 to SH and HQ). The funders had no role in study design, data collection and analysis, decision to publish, or preparation of the manuscript.

"double-edged-sword" effect (conflict between killing and strengthening by treatment) establishes an intrinsic limit to all cell-killing therapies. To optimize therapy a mathematical framework considering key quantitative parameters of treatment is necessary to predict which way the double-edged-sword will cut. Here we present an analytical model that define the parameter regimes in which tumor eradication either can or fundamentally cannot be achieved, but containment can be maximized.

## Introduction

The single major cause of treatment failure in cancer therapy is the emergence of a treatment-resistant tumor that drives recurrence. Other than in the case of some early-stage tumors, it is tacitly accepted that relapse is inevitable during the course of drug treatment. This assumption has translated into the unquestioned practice of quantifying the efficacy of all treatments by how long one extends the time period between diagnosis and either progression, in the form of a clinical relapse, or death [1]. The former metric defines a progression-free survival (PFS) time, which quantifies not the prevention of progression, but a delay, as evident in the proportional hazard model [2]. Treatment success is therefore measured as a "shift to the right" of the decaying Kaplan-Meier curve, which represents the fraction of progression-free surviving patients in the treated cohort compared to that of the control group. The hence derived extension of median PFS time (or loosely equivalent, of the median time to progression, or TTP) has become a de facto measure for success of a therapy [3].

In theoretical and in vitro experimental models of post-treatment regrowth of a tumor cell population, the "recovery time" in which the surviving tumor cells regrow to reach the population threshold present at the onset of treatment is a biological characteristic of the tissue. It can be considered the equivalent of the clinical TTP [4].

Recurrence, or tumor cell regrowth after treatment, is generally thought to be the result of selection in a process of Darwinian somatic evolution: Given sufficient genetic variability in a sufficiently large initial (pretreatment) cell population, it is considered statistically certain (possibly as a result of increased mutation rate in cancer cells) that the population contains cells carrying genomic mutations that confer drug-resistance and stem-like traits [5–8]. A single cell with such a mutation will survive the treatment and clonally expand, thus driving the tumor regrowth under treatment.

This genetic explanation implicitly acknowledges de facto inevitability of relapse for a certain set of parameters including mutation rate, cell population size and selection pressure [5, 6, 8, 9]. In addition, elimination of drug-sensitive cells by treatment releases drug-resistant cells from spatial and nutritional constraints and facilitates their proliferation, thereby creating an apparent causal link between treatment and recurrence [4, 10–13].

In recent years, non-genetic cell phenotype plasticity and the resulting cell population heterogeneity has been recognized as a source of the resistant cell phenotype, which could underlie recurrence without implication of genetic mutations [8, 14–19]. Extensive phenotypic heterogeneity within a population of cells is generated by the variability of an individual cell's biochemical state. Such non-genetic variability emanates, in part, from the ability of the cell's gene regulatory network to produce a diversity of stable gene expression patterns (attractors), resulting in multistability within a single clonal, isogenic population. Gene expression noise stochastically disperses individual cells in gene expression space, allowing them to occupy a range of these stable cell states. Because such stochasticity of gene expression causes

continuous phenotype switching and equilibration of phenotypes, this type of heterogeneity is not subject to the rule of extinction of a phenotype, as is the case with genetic mutation.

The resulting phenotypic diversity of the isogenic cell population is, while also probabilistic, more prevalent than that generated by genetic mutations and it produces distinct, robust, recurrent, and biologically relevant phenotypic sub-states in clonal cell populations [20–22]. Such sub-states include mesenchymal, persister, or stem-like states, etc., as single-cell RNAseq now amply reveals [19, 23–30]. Some of these states can confer resistance and are sufficiently robust to be inherited across several cell generations [22, 31]. In this manner, non-genetic probabilistic variation can also drive Darwinian selection of resistant cells, at least for a number of cell generations.

While both genetic and non-genetic variation arise in a probabilistic manner, there is a key difference. Because non-genetic variant attractor states are the result of regulatory mechanisms, they can also be directly induced by environmental signals. Such *instruction* to change gene expression programs in a directed manner by an external input via a deterministic (or strongly biased probabilistic) control, as opposed to *selection* of an undirected probabilistic internal change, plays a dominant role in a tumor's acquisition of stem-like drug-resistant cells at short time scales (days) compared to the clonal expansion of rare mutant clones [9]. Such regulated change of cell state may be part of a robust, evolved cellular defense mechanism against cellular toxins [32].

A growing body of evidence suggests that emergence of stem-like and therapy-resistant cells along with the associated changes in gene expression are induced (as opposed to selected) by the cytotoxic stress of treatment [29, 33, 34]. In other words, there is a causal biological mechanism linking treatment to stress to the stem-like phenotype. The recurring appearance of stem-cell-like gene expression programs, or "stemness," the speed of response and involvement of canonical signaling pathways that confer multidrug resistance (such as Wnt signaling-mediated upregulation of the ABC membrane pumps) collectively support the idea of stress-induced activation of preexisting xenobiotic resistance programs in cells by treatment [35–37].

Therefore, any cytocidal treatment may be a double-edge sword: while a one fraction of cells in the heterogeneous population is killed, another fraction of cells is induced by treatment stress to enter a stem-like or more resistant persister state—planting the seed for recurrence [8, 38]. Drug-induced resistance thus poses an intrinsic limit to curability of tumors under any treatment that involves cell stress, as is the case with chemotherapy, target-selected therapy or radiation. The role of somatic Darwinian evolution in recurrence relative to that of non-genetic cell state transitions has been analyzed using mathematical models in order to minimize selection pressure during treatment [4, 5, 39–41]. However, the intrinsic limit that induced resistance places on therapy has only recently been systematically evaluated [39].

Here we analyze a most elementary, generic, mathematical model for the conflicting processes that are inherent to cancer therapy: treatment-induced cell death and treatment-induced transition from a drug-sensitive phenotype to a drug-resistant (stem-like) one. Under this formulation, recurrence is "wired-into" the population dynamics and, depending on quantitative details, can become manifest. Over a relevant parameter space of an ordinary differential equation (ODE) model, we analytically evaluate the activity profiles of a drug in inducing cell death vs. transition to the resistant state. We quantify how these features of treatment relate to the intrinsic inevitability of recurrence, measured as TTP. We thus provide a formal survey of the consequence of the core process of non-genetic induction of resistance by treatment, irrespective of the ensuing selection and micro-environmental influences.

We find that depending on the relative susceptibilities for cell death versus induction of the resistant state there can be a non-monotonic dependence of TTP on drug dose, such that beyond a narrow optimal dose, the induction of resistance dominates and increasing treatment

intensity will drastically shorten TTP. Thus, knowledge of the measurable propensities of cells to die or activate resistance mechanisms as a function of dose is critical information for optimizing therapy. Our focus on treatment-induced non-genetic tumor cell state change complements the evolutionary framework, fills a conceptual gap to help explain why it is so hard to cure advanced cancer and can be used for modeling scheduling to avoid treatment-associated progression as sought by adaptive therapy [11].

## Materials and methods

### Dynamical model of tumor growth

We consider an ODE model that describes the population dynamics of cancer cells that interconvert between two distinct cell states: drug-sensitive and drug-resistant. Let $x(t) = [x_1(t), x_2(t)]^T$ denote the population state vector, where $x_1(t)$ is the number of sensitive cells, and $x_2(t)$ is the number of resistant cells. Following the formulation given by Zhou et al., the sizes of these two populations are governed by the following linear ODE (Fig 1) [42]:

$$\frac{dx}{dt} = Ax, \quad A = \begin{bmatrix} b_S - d_S - k_{SR} & k_{RS} \\ k_{SR} & b_R - d_R - k_{RS} \end{bmatrix} \tag{1}$$

We analyze this model of ongoing treatment with the assumption that treatment acts by raising the per capita death rate of cancer cells. Herein, the parameter $b_S$ ($b_R$) denotes the fixed division rate of sensitive (resistant) cells, whereas $d_S$ ($d_R$) denotes the total death of sensitive (resistant) cells undergoing drug treatment. The parameter $k_{SR}$ denotes the transition rate at which sensitive cells become resistant, and $k_{RS}$ is the transition rate at which resistant cells become sensitive. All of these parameters are strictly positive and depend in a particular way from the drug dose as discussed below.

We use TTP—the time it takes for the tumor to surpass its pretreatment population size—as a quantitative measure of recurrence under drug treatment. In this model, we denote TTP by $t_P$, which is defined as

$$t_P = \inf\{t > 0 \,|\, N(t) > N(0)\} \tag{2}$$

where $N(t) = x_1(t) + x_2(t)$ is the total number of cells in the tumor (Fig 1). The goal of this model is to understand how $t_P$ depends on the model parameters.

**Stability analysis.** Aside from the degenerate case in which one eigenvalue of $A$ is zero, the only fixed point of the ODE in Eq 1 is the point $x = 0$: extinction of all cancer cells. The eigenvalues $\lambda_{1,2}$ of $A$ determine the local stability of this fixed point, whereas the eigenvectors $v^{(1,2)}$ of $A$ give the primary directions of growth and/or decay in state space. For this model, we can derive general conditions on the growth and transition rates under which the origin is an unstable node, a stable node, or a saddle point (Table A in S1 Text). Under appropriate initial conditions, these three cases correspond to unchecked tumor growth, tumor extinction, and tumor regression followed by regrowth. The last scenario provides the simplest mathematical conception for the relapse phenomenon.

The saddle point nature of the extinction state suggests that the dynamics are ultimately difficult to control and contain. It also points to a marked "turning point" for each trajectory: if the initial tumor population is "close enough" to the stable manifold, the trajectory will first move towards the origin before being repelled away along the unstable manifold, indicating the difficulty to eradicate the tumor (S1–S4 Figs). We can think of this behavior as temporary

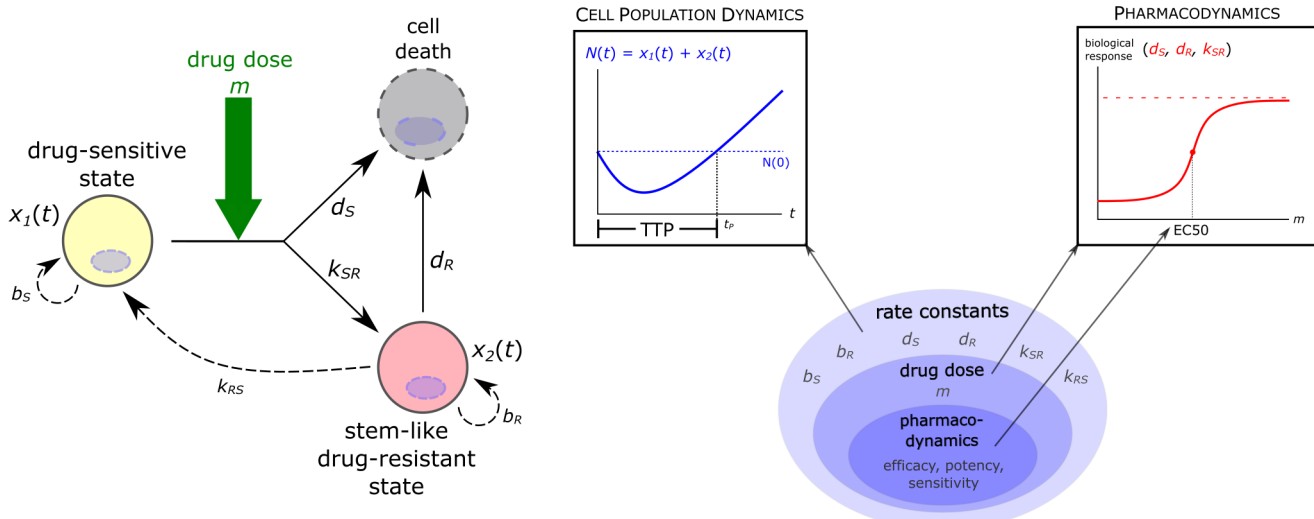

**Fig 1. Levels of parameterization in the dynamical model of tumor growth.** At the "highest" (i.e., most general) level, there are the rate constants that govern the growth and state transition dynamics of the cancer cell population. One level down, we introduce drug treatment into the model by assuming that the rate constants $d_S$, $d_R$, and $k_{SR}$ are logistic functions of drug dose $m$. The parameters that determine the shape of these dose-response curves—the pharmacodynamic constants, e.g., EC50—form the "lowest" (i.e., most specific) level of the model parameters.

control of tumor size before the inevitable progression. S3 Eq in S1 Text tells us that it is possible to control tumor regrowth wherever the relative fitness of the sensitive phenotype is sufficiently large.

The extinction state is a saddle point whenever the ratio $(b_S - d_S)/(b_R - d_R)$ is greater than the ratio $\frac{(b_S - d_S - k_{SR})}{k_{RS}}$ (S3 Eq in S1 Text). That is, temporary control of the tumor size is possible when the relative fitness of the sensitive phenotype is sufficiently large. The constant $b_S - d_S - k_{SR}$ is the net flux of the drug-sensitive population per unit density of drug-sensitive cells, and the constant $k_{RS}$ is the flux into the drug-sensitive population per unit density of drug-resistant cells. The latter parameter, sometimes referred to as the "backflow rate," is useful in characterizing how the pool of drug-resistant cells allows the drug-sensitive population to avoid extinction during chemotherapy [42].

**Time to progression.** The behavior of TTP as a function of drug dose is closely related to the saddle point dynamics of the system. For one, to have $t_P > 0$, the total population must decrease initially before recovering to its initial value, which is typically only observed when the origin is a saddle point. Moreover, under the saddle point dynamics, $t = t_P$ is the unique non-zero time point for which $N(t) = N(0)$.

Even with a closed expression for $N(t)$, solving the expression $N(t_P) = N(0)$ for $t_P$ is not generally possible as it involves the sum of distinct exponential terms. Instead, we can approximate $t_P$ by decomposing the saddle point dynamics into the distinct stages of population decrease (remission) and increase (regrowth). By definition of TTP, the total population $N(t)$ is undergoing regrowth at time $t = t_P$, i.e., $N'(t_P) > 0$. The theory of linear dynamical systems tells us that the exact solution $N(t)$ is given by a linear combination of the exponential terms $e^{\lambda_1 t}$ and $e^{\lambda_2 t}$, where $\lambda_{1,2}$ are the eigenvalues of $A$ (S5 Eq in S1 Text). In the case of a saddle point, where $\lambda_1 > 0$ and $\lambda_2 < 0$, the tumor population regrowth is necessarily driven by the exponential term corresponding to the positive eigenvalue $\lambda_1$. Therefore, at time $t_P$, we may assume that the total population is well approximated by this exponential term: $N(t_P) \sim e^{\lambda_1 t_P}$. Under this

assumption, we can approximate TTP as follows (S8 Eq in S1 Text):

$$t_P \approx t_P^* := \frac{1}{\lambda_1} \ln \left[ \frac{N(0)}{c_1 (v_1^{(1)} + v_2^{(1)})} \right] \tag{3}$$

## Pharmacodynamic model of continuous therapy

To consider drug treatment, we assume that drug dose is constant throughout the course of therapy (i.e., continuous therapy) and that the rate constants $d_S$, $d_R$, and $k_{SR}$ depend on the amount of drug $m$ present in the system, i.e., that chemotherapy reduces tumor burden by increasing the death rate of cancer cells. At the same time, it increases the rate at which sensitive cells become resistant, the basis for the "double-edged sword" effect of chemotherapy. In mathematical terms, we introduce a secondary parameter $m$ that denotes the drug dose, and we assume that the primary parameters $d_S$, $d_R$, and $k_{SR}$ that capture tumor cell population dynamics are increasing functions of $m$ (Fig 1). Scaling $m$ as percentage of the maximum tolerated dose (MTD), i.e., $0 \leqslant m \leqslant 100$, we next discuss the pharmacodynamics of cellular drug response, i.e., functional forms for the dependence of the three primary parameters from $m$.

Using the commonly observed sigmoid shape of biological response curves, which reflect the cumulative probabilistic response of individual cells in a heterogeneous population, we use logistic functions to describe the rate constants $d_S(m)$, $d_R(m)$, and $k_{SR}(m)$ (Fig 1):

$$\begin{cases} d_S(m) = \delta_S + \frac{E_S}{1+\exp(-r_S(m-P_S))} \\ d_R(m) = \delta_R + \frac{E_R}{1+\exp(-r_R(m-P_R))} \\ k_{SR}(m) = \kappa + \frac{E_{SR}}{1+\exp(-r_{SR}(m-P_{SR}))} \end{cases} \tag{4}$$

We assume that $k_{RS}$, the rate constant for the re-sensitization of resistant cells, does not change with drug dose.

In the case of $d_S(m)$, each of the four parameters $\delta_S$, $E_S$, $r_S$ and $P_S$ determines the shape of the logistic curve and describes a behavior affected by the drug. For this reason, we refer to these parameters as the *pharmacodynamic parameters* associated with a given rate constant for cell response. In particular, the parameters $E$, $P$, and $r$ respectively determine the maximum response (*efficacy*), EC50 (which is inversely related to *potency*), and saturation rates for each of the above responses of the drug. For example, a drug with high efficacy and high potency to kill sensitive cells is characterized by a high value for $E_S$ and a low value for $P_S$.

Taken together, our model incorporates three levels of parameterization: first, the parameters of the general linear population dynamics model are the rate constants $b_S$, $b_R$, etc. (Fig 1). Second, some of these rate constants (e.g., $k_{SR}$) depend on drug dose, which is represented by the parameter $m$ (Fig 1). And third, the way each of these rate constants depend on drug dose is determined by their respective pharmacodynamic constants, as given in Eq 4 (Fig 1).

In drug development implicit pharmacodynamic parameters are empirically tuned to optimize treatment outcome, which, in the case of our model, is $t_P$. Teasing apart how the drug dose $m$ affects TTP as a function of the pharmacodynamic parameters is not a straightforward problem because we do not have an explicit expression for $t_P$. An exhaustive search of the entire parameter space is impractical. Because drug-induced transition to the resistant state is a hitherto unaccounted-for factor affecting recurrence, we start with a cursory analysis of how $t_P$ changes when we vary the dose-response relationship of the sensitive-to-resistant transition rate $k_{SR}(m)$.

## Results

### Qualitative sensitivity analysis

Representative samples of the parameter space demonstrate the possible qualitatively distinct treatment outcomes—$t_P$ as a function of drug dose—that depend on the pharmacodynamics (i.e., rate of behavior as function of drug dose $m$) of drug-induced resistance relative to killing (Fig 2). We define "drug resistance" by taking $E_R \ll E_S$: the drug's cell-killing efficacy is lower for the drug-resistant phenotype than it is for the drug-sensitive phenotype. Although drug-resistance can be manifest in dampening the killing effect either by decreasing the drug potency (or equivalently, increasing the EC50) or efficacy to kill cells (or a combination of both), we do not vary this aspect of resistance. Instead, we model resistance as a lowering the resistant cell death rate at MTD, which reflects a more profound effect on the cell state. To incorporate the observed fitness cost of resistance in the absence of drug, we further assume that $b_S > b_R$ and $\delta_S = \delta_R$ [4, 10, 43, 44].

Thus, we anchor the pharmacodynamics for killing of sensitive and resistant cells $d_S(m)$ and $d_R(m)$ and vary the parameters that determine drug-induced transition to the resistant state with respect to either EC50 ($P_{SR}$) or efficacy ($E_{SR}$) for the transition rate $k_{SR}(m)$ (S1 Table, Fig 2). We consider these two parameters specifically because of the different effects that they exert on the dose-response curve of a given drug. The efficacy of a drug, or the drug response at MTD, is an intrinsic property of the drug and cannot be compensated by alteringg the drug dose. On the other hand, the potency of a drug, which corresponse to the inverse of the EC50, is a property that can be compensated by altering drug dose: a low-potency drug can achieve the same response as a high-potency drug, but at a higher drug dose.

The unknown then is how the relation between rate of induction of the resistant phenotype, $k_{SR}(m)$, and the "kill curves," $d_S(m)$ and $d_S(m)$, shape $t_P$ as a function of drug dose $m$. The coarse-grained, but comprehensive, sensitivity analysis of our model model is achieved by

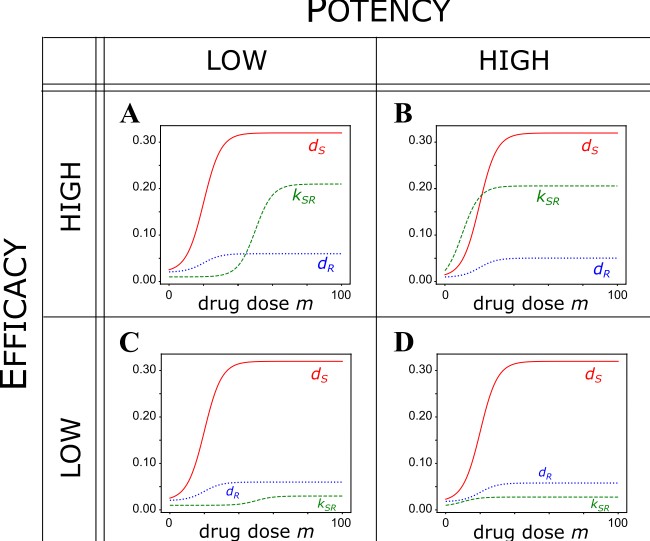

**Fig 2. A coarse-grained sensitivity analysis of the effect of drug-induced resistance on treatment outcome.** The parameters given in S1 Table are held fixed, while the EC50 and efficacy of resistance induction are varied in the above four cases. Efficacy increases as the parameter $E_{SR}$ increases, whereas potency decreases as the EC50 value ($P_{SR}$) increases. **A**. Case A: high efficacy, low potency ($E_{SR} = 0.2$, $P_{SR} = 50$). **B**. Case B: high efficacy, high potency ($E_{SR} = 0.2$, $P_{SR} = 10$). **C**. Case C: low efficacy, low potency ($E_{SR} = 0.02$, $P_{SR} = 50$). **D**. Case D: low efficacy, high potency ($E_{SR} = 0.02$, $P_{SR} = 10$).

altering the values of $P_{SR}$ and $E_{SR}$, relative to the two fixed kill curves pharmacodynamics curves (Fig 2). The following four canonical cases represent qualitatively distinct, plausible pharmacodynamical relationships between killing and induction of resistance due to treatment.

**Case A: High efficacy, low potency of resistance induction.** We first consider a drug with a high efficacy and low potency (i.e., high $E_{SR}$ and $P_{SR}$) for inducing resistance (Fig 2A). For such a drug, $t_P$ is not monotone increasing in drug dose. Instead, it first increases before decreasing to a plateau (Fig 3A). Thus, increasing drug dose past a certain point significantly worsens the treatment outcome.

We observe that $t_P$ increases with drug dose when an increase in dose corresponds to a significant decrease in the total growth rate relative to the total switching rate (Fig 3A). On the other hand, $t_P$ decreases with drug dose when an increase in drug dose corresponds to a significant increase in the total switching rate relative to the total growth rate (Fig 3A). The fact that the transition dynamics dominate the growth dynamics at high drug doses indicates that under cytotoxic stress, tumor recurrence is not driven by cell growth (Fig 3). Instead, it is driven by the ability of the tumor cells to evade extinction by transitioning to a stem-like, drug-resistant state.

Returning to our analysis of the saddle point dynamics, the positive eigenvalue $\lambda_1$ is also non-monotonic in drug dose, first decreasing to a minimum before increasing to a plateau (Fig 3). This inverse relationship between $\lambda_1$ and $t_P$ agrees with the above observation that $\lambda_1$ roughly corresponds to the rate of tumor growth during recurrence.

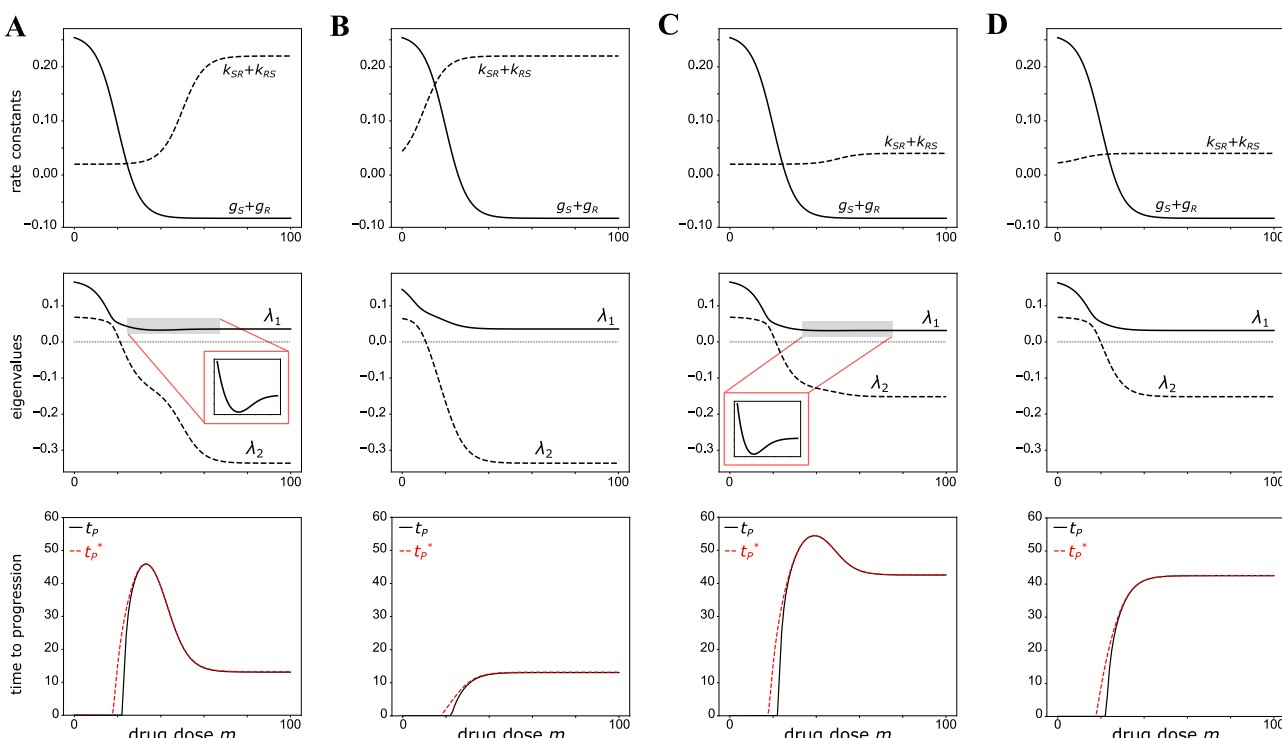

**Fig 3. Summary of model behavior for Cases A, B, C, and D as a function of drug dose $m$.** The net growth rates of sensitive and resistant cells are defined as $g_S := b_S - d_S$ and $g_R := b_R - d_R$, respectively. *First row*: Total growth and transition rates, $g_S + g_R$ and $k_{SR} + k_{RS}$, respectively. *Second row*: Eigenvalues of the matrix $A$ in Eq 1. The inset in cases A and C highlight the non-monotonic behavior of the eigenvalue $\lambda_1$. *Third row*: Time to progression $t_P$ plotted alongside its asymptotic approximation $t_P^*$.

**Case B: High efficacy, high potency of resistance induction.** We now consider a drug with the same efficacy in inducing the stem-like state as in Case A but with higher potency (Fig 2B). Unlike Case A, $t_P$ is now monotone increasing in drug dose (Fig 3B). Therefore, if the drug dose is sufficiently high, the treatment outcome in Case B is not sensitive to fluctuations about a dosage. This robustness is favorable to the sensitivity observed under Case A, in which a small fluctuation in drug dose can significantly worsen the $t_P$. On the other hand, the maximum $t_P$ in Case B is low compared to that in Case A (Fig 3A and 3B). A treatment that is robust to variation in drug dose is not necessarily a good one if progression is only, at most, slightly delayed.

As before, the behavior of $t_P$ is well summarized by the positive eigenvalue $\lambda_1$ and the difference between the total growth and switching rates (Fig 3B). In particular, the latter shows how increased drug potency to induce stemness affects $t_P$. As EC50 for $k_{SR}$ is closer to that for $d_S$ and $d_R$ than it is in Case A, the total growth and state-switching rates saturate at roughly the same drug dose (Fig 3B). Therefore, the difference between the two rates only increases over a small range of doses, in which in $t_P$ increases monotonically (Fig 3B). Cases A and B reveal that the relative potency of a given drug to induce resistance versus kill cells determines whether $t_P$ is monotone increasing in drug dose or not.

**Case C: Low efficacy, low potency of resistance induction.** For a drug with the same low potency as in Case A, but with lower efficacy, $t_P$ is again, as in Case A, a non-monotonic function of drug dose (Figs 2C and 3C). However, the difference between the maximum possible $t_P$ and $t_P$ at MTD is much smaller for the drug than in Case A, since the maximum difference between the total growth and switching rates is smaller than in Case A (Fig 3A and 3C). In terms of treatment outcomes, we can think of this drug as an improvement from Cases A and B: $t_P$ is overall less sensitive to variation in drug dose than in Case A, and the maximum possible $t_P$ is greater than that in both Cases A and B.

The overall increase in $t_P$ is also reflected in saddle point dynamics of the system. In Case C, the magnitude of the negative eigenvalue $\lambda_2$ is overall lower than in Case A (Fig 3C). This decrease in magnitude agrees with the increase in $t_P$, as $\lambda_2$ roughly corresponds to the rate of tumor remission in the early stages of treatment. That is, the remission stage is prolonged under treatment by a low-efficacy drug, as compared to a high-efficacy drug (S1–S4 Figs).

**Case D: Low efficacy, high potency of resistance induction.** Finally, we consider a drug that is a "combination" of Cases B and C; that is, the drug has low efficacy but high potency in inducing resistance to cell killing (Fig 2D). Under this drug, $t_P$ is a similar "combination" of Cases B and C: it is monotone increasing in drug dose, and its value at MTD is the same as in Case C (Fig 3D). Therefore, a drug with low efficacy and high potency for drug-induced resistance relative to killing produces a treatment scheme that effectively delays tumor progression for a wide range of drug doses. Compared to case C, the higher potency in inducing stemness at low dose abrogates the optimal dose window, i.e., the counterintuitive peak in $t_P$ at lower dose.

## Parameter search

In order to determine how well Cases A, B, C, and D capture the dependence of $t_P$ on drug potency ($P_{SR}$) and efficacy ($E_{SR}$) of resistance induction, we perform a broad parameter search. Specifically, we compute $t_P$ over a wide range of $P_{SR}$ and $E_{SR}$ beyond the four representative cases from the previous section (Fig 2). Instead of plotting $t_P$ as a function of drug dose $m$, we plot three quantities of interest as a function of $P_{SR}$ and $E_{SR}$: the drug dose at which $t_P$ attains its maximum, the maximum value of $t_P$, and the value of $t_P$ at MTD (Fig 4). Doing so, we find that $t_P$ attains its maximum at MTD (i.e., is monotonic in drug dose) when $P_{SR}$ is low

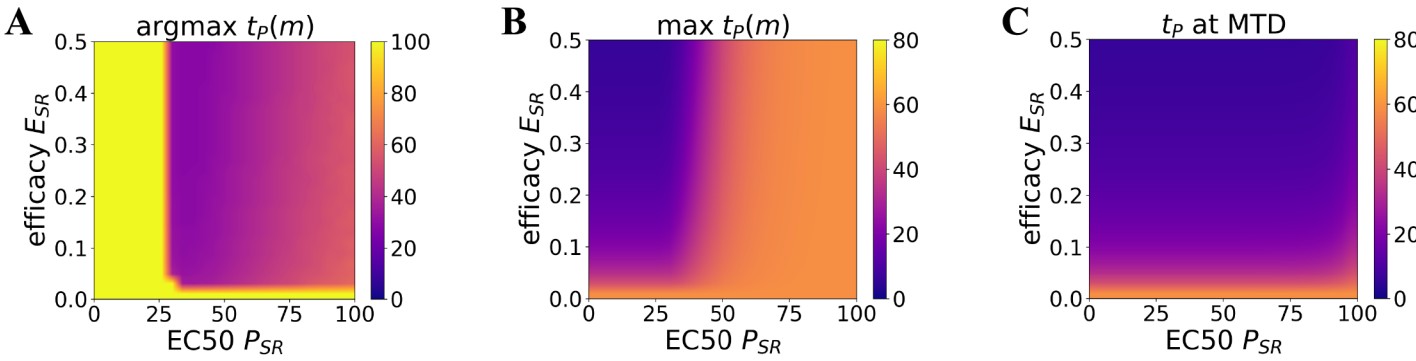

**Fig 4. Analysis of time to progression ($t_P$) over a wide range of values for potency ($P_{SR}$) and efficacy ($E_{SR}$) of inducing resistance.** Plotted values are indicated by the colorbar on each plot. All other parameters are fixed at the values given in S1 Table. **A**. Drug dose $m$ at which $t_P$ attains its maximum. **B**. Maximum value of $t_P$ over all drug doses. **C**. Value of $t_P$ at MTD.

regardless of the value of $E_{SR}$ (Fig 4A). Therefore, $t_P$ is indeed a non-monotonic function of drug dose only when the potency to induce resistance is low. Furthermore, the maximum value of $t_P$ shows little sensitivity to either $E_{SR}$ or $P_{SR}$, and instead appears to depend on whether $t_P$ is monotone increasing in drug dose or not (Fig 4B). We also find that the value of $t_P$ at MTD decreases as $E_{SR}$ increases independent of the value of $P_{SR}$ (Fig 4C).

The results of this finer-grained parameter search indicate that the four cases given in the main text indeed characterize the dependence of $t_P$ on the parameters $P_{SR}$ and $E_{SR}$: the non-monotonic dependence of $t_P$ on drug dose is governed by $P_{SR}$, whereas the value of $t_P$ at MTD is governed by $E_{SR}$. This parameter search, however, is still limited to the two parameters $P_{SR}$ and $E_{SR}$. Taken together, a key finding is that at low potency (high EC50) for inducing resistance, where higher drug doses $m$ are needed to trigger this cell state transition, increasing the dose past a certain point reduces $t_P$. A further evaluation of the parameter space, paired with rigorous sensitivity analysis, is required to characterize how $t_P$ depends on the complete set of model parameters.

## Virtual cohort simulations

In statistical analysis of clinical studies, individual patient measures, such as PFS time and TTP, are typically not displayed directly; instead, the data are often presented in the form of Kaplan-Meier curves, which show the fraction of surviving and progression-free patients as a function of time [2, 3]. The process of calibrating mathematical models of cancer dynamics to clinical data typically involves applying regression, or some other parameter-fitting method, to a Kaplan-Meier curve from observed cancer patient cohorts [45, 46]. Doing so requires generating a "virtual patient cohort" from the model, usually by assuming some statistical distribution of a set of the model parameters, and sampling individual "patients" from this distribution [45, 46]. As a step towards grounding our model in clinical data, so as to make meaningful predictions about treatment courses, we generate virtual patient cohorts and present an analysis of the resulting Kaplan-Meier curves.

To generate virtual patient cohorts, we assume that the basal rates of cell birth, death, and state transition, as well as the fraction of resistant cells at tumor detection, vary from patient to patient (S2 Table). For simplicity, we assume that each of these parameters are uniform random variables. One possible way of updating this assumed prior with a more appropriate posterior distribution based on clinical data would be to use an expectation-maximization approach [46]. We assume that the remaining parameters, tumor size at diagnosis and the

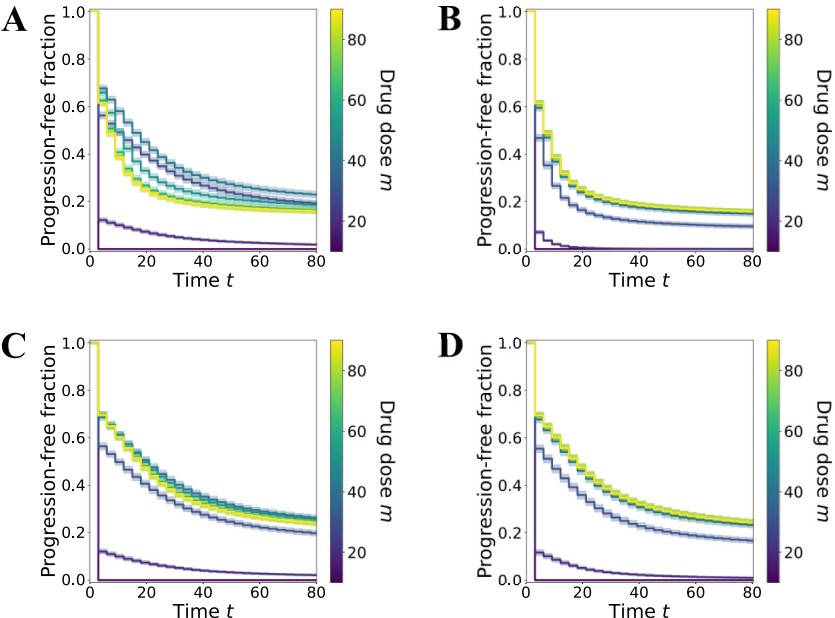

**Fig 5. Kaplan-Meier curves averaged over $n_c = 100$ virtual cohort simulations of $n_p = 10^3$ patients each, plotted for fixed drug doses $m$.** Error bars denote plus and minus one standard deviation. The $x$-axis denotes time $t$, and the $y$-axis denotes the fraction of patients in a cohort who are progression free by time $t$ (i.e., $t_P > t$). Each curve is colored according to the fixed drug dose $m$ applied to each cohort, indicated by the color bar in each plot. Panels **A**, **B**, **C**, and **D** correspond to varying the parameters $P_{SR}$ and $E_{SR}$ according to the four cases shown in Fig 2. Unless specified as being drawn from the uniform distributions given in S2 Table, all other parameters are held fixed at the values given in S1 Table.

pharmacodynamic constants, remain fixed across all patients, as they are intrinsic properties of standard clinical practice and a given drug, respectively (S1 Table).

Keeping with our previous analysis, we generate $n_c = 100$ virtual patient cohorts of $n_p = 10^3$ patients each, and compute the TTP curve $t_P(m)$ for each patient at values of $E_{SR}$ and $P_{SR}$ given by the four previous cases A, B, C and D (Fig 2). Once we have $t_P(m)$ for each patient in a given cohort, we can compute the progression-free fraction for a range of possible drug doses, where we treat $t_P$ as a progression event for each patient. We then take the mean progression-free fraction across all cohorts, giving us a set of Kaplan-Meier curves (Fig 5). Our aim is to see if the non-monotonic dependence of treatment outcome on drug dose $m$ is manifest at the ensemble level of the patient cohort. Whereas for an individual patient, an improved treatment outcome is indicated by an increased TTP, an improved outcome for an entire cohort is indicated by an upward and/or rightward shift in the Kaplan-Meier curve.

Comparing the resulting Kaplan-Meier curves for cases A-D, we find that the mean behavior of the survival fraction mirrors that of $t_P$. As before, in cases A and C, where potency to induce resistance is low, the overall treatment outcome is non-monotonic in drug dose: as $m$ increases from 0 to MTD, the Kaplan-Meier curve shift upwards, then downwards (Fig 5). Also in line with our TTP analysis is the observation that in cases C and D, where efficacy to induce resistance is low, the overall treatment outcome is better than that for cases A or B: the upwards shift of the Kaplan-Meier curves going from low to high drug doses is much larger in the former cases than in the latter (Fig 5). Thus, our conclusions about the qualitative behavior of the model at the level of the individual "patient's" tumor cell population scale up to the level of the statistical ensemble (i.e., the cohort), replicating both the "expected" as well as the "counterintuive" effects of increasing drug dose. This result is significant as it demonstrates

that our analysis is in some sense robust to noise. The next logical step would be to see if our analysis still holds, and is therefore clinically meaningful, after fitting our parameters to actual patient data.

## Discussion and conclusion

In advanced tumors, post-treatment recurrence is almost an intrinsic feature in the course of tumor progression. It is increasingly acknowledged that even if an untreated tumor would result in rapid progression and death, recurrence after treatment is causatively or mechanistically linked to the act of treatment. The traditional explanation for recurrence after initial remission invokes selection of preexisting mutant cells in which genetic mutations confer the resistance phenotype.

More recently, the "competitive release" of the resistant cells, when these cells expand into the niche freed by the killing of sensitive cells by treatment, has been proposed as mechanism of tumor recurrence, adding the non-intuitive twist that "more killing" is not better [4, 10–13, 39]. Nonlinear models of competition between sensitive and resistant subpopulations, such as that presented by Kozlowska *et al.*, lead to similar saddle-point dynamics of tumor depletion and progression as our model, albeit at a longer time scale that encompasses multiple cell generations [46].

Another interesting fundamental similarity between genetic and non-genetic mechanisms, at least in terms of a possible formal generalization, pertains to an additional layer of non-linearity that we have not considered here: a multi-step process that gives rise to a progressive increase ('gradual evolution') in resilience. Increasing genomic instability with tumor progression not only decreases the threshold for cell death exploited by many therapies, but also accelerates mutational exploration of new phenotypes in an evolutionary process. A well-studied case of self-propelling increase in resistance is the amplification of the DHFR gene repeats that confers resistance to methotrexate. Once amplified, the locus is even more prone to undergo genomic recombinations and to further amplify, including in response to treatment stress that promotes chromosomal breakage [47]. The non-genetic equivalent is that with increasing malignancy ("stemness"), the barrier for cell type transitions is reduced, evident in the increasing phenotypic plasticity of advanced tumors, and hence, the chance of the tumor cell to enter an even more dedifferentiated, resilient stem-like cells in response to treatment stress [47].

The oft observed re-sensitization of recurrent tumors, rapid rate for appearance of resistance markers, and ubiquitous cell phenotype plasticity, however, suggest a role for non-genetic, reversible phenotype switching in tumor recurrence [14–19, 48–50]. Most recently, the induction of cells to transition into a stem- or mesenchymal-like resilient state by the cell stress imparted by treatment has received increasing acceptance and has been confirmed by single-cell resolution measurements [19–30]. Population-wide resistance to treatment induced by the same perturbation intended to kill and eradicate tumor cells thus poses a conflicting situation: a double-edged sword that complicates treatment response.

The mechanisms that allow a cell to be either fated to death or to enter a resistant state following the same perturbation may depend on the initial microstate due to stochastic gene expression [38]. This uncertainty is here modeled by the sigmoidal shape of the pharmacodynamical functions, which represents the cumulative probability of the dispersed propensity of cells to respond in either way to treatment. We used a minimal model to characterize how the "relative strength" (potency and efficacy) of a drug to either kill tumor cells or convert them into resistance cells affects the population dynamics, as measured by the time it takes for the cell population to recover and grow to its pre-treatment size (time to progression $t_P$). We focused on four qualitatively distinct scenarios (A, B, C, and D) that correspond to all possible

combinations of high versus low potency and high versus low efficacy of the treatment to induce the resistant state relative to killing.

Despite the elementary form of the model, interesting behaviors emerge: the four scenarios produced robust, prototypic behaviors for the dependence of $t_P$ on drug dose (Fig 4). The two cases (A and C) in which the potency of the drug (dose for half-maximal effect, EC50) to induce the resistant phenotype was substantially lower (high EC values) than the potency to kill cells produced a non-monotonic $t_P$-to-drug dose relationship. In such a scenario, there is an optimal window of dose for maximal $t_P$: drug doses higher than the optimal dose will have lower benefits in terms of $t_P$.

The efficacy of the drug (the amplitude of the dose-response curve) also affected $t_P$ by determining its plateau value at high drug doses. High efficacy of the drug to induce the resistant state (Cases A, B) resulted, as expected, in a low value of $t_P$ at high doses, independent of whether the dose-dependence is monotonic (Case B) or non-monotonic (Case A). On the other hand, a drug with low efficacy to induce treatment resistance (Cases C, D) resulted in a high value of $t_P$ at high doses. Sensitivity of $t_P$ to changes in drug efficacy means that drug dose optimization is paramount in the case where the potency of the drug to induce resistance is low relative to the potency of cell killing—parameters that could be determined in preclinical studies.

Unfortunately, fine-grained dose-escalation clinical trials (e.g. with at least three doses) are generally not conducted that would expose the non-monotonic effect. To establish the connection between intrinsic biological properties of drugs in triggering state transition to the resistant state and the clinical consequences as observed in drug trials, we performed virtual patient cohort simulations. We found that the qualitative conclusions of the model about the effect of induced resistance on treatment success are robust to variation in the other model parameters and result in corresponding non-monotonic dependence of the progression free survival in the simulated patient drug trial cohorts.

Adaptive therapy, in which treatment is stopped upon regression and re-started upon regrowth, or metronomic therapy, which applies a low dose at regular intervals, may be worthwhile treatment schemes in cases A and C because there is an optimal drug dose below MTD in these cases. The current rationale behind dose-minimization treatment strategies is to avoid fixation of mutant resistant clones due to competitive release and selection [10–13, 40]. However, if resistance is inducible and reversible, the intended "containment" of the tumor (as opposed to the harder-to-achieve "eradication") is even more readily achieved than when guided by the concept of competitive release of mutant resistant clones. Thus, if we consider the new biological rationale of non-genetic, reversible dynamics of treatment-induced resistance, a strategy such as adaptive or metronomic therapy may be further optimized to be more effective. However, in order to make any meaningful predictions about optimal treatment courses we must first ground our model in clinical data, either by using empirical estimates of parameters from the literature, or by using a statistical learning framework to fit parameter distributions from the data [45, 46, 51, 52].

## Supporting information

**S1 Text. Table A. Summary of all possible cases for the stability of the origin under the ODE in Eq 1.**
(ZIP)

**S1 Fig. Dynamics of tumor population recovery for Case A. A**. *Left*: Time-evolution of the total population $N(t)$ plotted on a logarithmic scale for drug doses $m$ = 23, 35, 45, 85. The horizontal dashed line indicates initial population size $N(0)$. *Right*: Turning point $t_{min}$, at which

$N(t)$ reaches a minimum, as a function of drug dose $m$. Red markers indicate reference points of $m = 23, 35, 45, 85$. **B**. Parametric curve $(t_{min}(m), t_P(m))$ relating the turning point $t_{min}$ and the time to progression $t_P$. Dashed lines of slope 2 (blue) and 3 (red) are given for reference. (EPS)

**S2 Fig. Dynamics of tumor population recovery for Case B. A**. *Left*: Time-evolution of the total population $N(t)$ plotted on a logarithmic scale for drug doses $m = 23, 35, 45, 85$. The horizontal dashed line indicates initial population size $N(0)$. *Right*: Turning point $t_{min}$, at which $N(t)$ reaches a minimum, as a function of drug dose $m$. Red markers indicate reference points of $m = 23, 35, 45, 85$. **B**. Parametric curve $(t_{min}(m), t_P(m))$ relating the turning point $t_{min}$ and the time to progression $t_P$. Dashed lines of slope 2 (blue) and 3 (red) are given for reference. (EPS)

**S3 Fig. Dynamics of tumor population recovery for Case C. A**. *Left*: Time-evolution of the total population $N(t)$ plotted on a logarithmic scale for drug doses $m = 23, 35, 45, 85$. The horizontal dashed line indicates initial population size $N(0)$. *Right*: Turning point $t_{min}$, at which $N(t)$ reaches a minimum, as a function of drug dose $m$. Red markers indicate reference points of $m = 23, 35, 45, 85$. **B**. Parametric curve $(t_{min}(m), t_P(m))$ relating the turning point $t_{min}$ and the time to progression $t_P$. Dashed lines of slope 2 (blue) and 3 (red) are given for reference. (EPS)

**S4 Fig. Dynamics of tumor population recovery for Case D. A**. *Left*: Time-evolution of the total population $N(t)$ plotted on a logarithmic scale for drug doses $m = 23, 35, 45, 85$. The horizontal dashed line indicates initial population size $N(0)$. *Right*: Turning point $t_{min}$, at which $N(t)$ reaches a minimum, as a function of drug dose $m$. Red markers indicate reference points of $m = 23, 35, 45, 85$. **B**. Parametric curve $(t_{min}(m), t_P(m))$ relating the turning point $t_{min}$ and the time to progression $t_P$. Dashed lines of slope 2 (blue) and 3 (red) are given for reference. (EPS)

**S1 Table. Initial conditions, rate constants and pharmacodynamic parameters with fixed values (units not specified).**
(XLSX)

**S2 Table. Parameter distributions used to general virtual patient cohorts.**
(XLSX)

## Acknowledgments

The authors wish to acknowledge Dr. M. Strasser and Dr. A. Brock for discussions that contributed to the development of this work.

## Author Contributions

**Conceptualization:** Joseph Xu Zhou, Hong Qian, Sui Huang.

**Data curation:** Erin Angelini.

**Formal analysis:** Erin Angelini, Yue Wang.

**Funding acquisition:** Hong Qian, Sui Huang.

**Methodology:** Yue Wang, Joseph Xu Zhou, Hong Qian, Sui Huang.

**Project administration:** Hong Qian, Sui Huang.

**Software:** Erin Angelini, Yue Wang.

**Supervision:** Hong Qian, Sui Huang.

**Visualization:** Erin Angelini.

**Writing – original draft:** Erin Angelini, Sui Huang.

**Writing – review & editing:** Erin Angelini, Yue Wang, Joseph Xu Zhou, Hong Qian, Sui Huang.

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
