## [Decision Letter · Decision Letter 0]

20 Jan 2022

Dear Prof. Huang,

Thank you very much for submitting your manuscript "A model for the intrinsic limit of cancer therapy: duality of treatment-induced cell death and treatment-induced stemness" for consideration at PLOS Computational Biology.

As with all papers reviewed by the journal, your manuscript was reviewed by members of the editorial board and by several independent reviewers. In light of the reviews (below this email), we would like to invite the resubmission of a significantly-revised version that takes into account the reviewers' comments.

Several reviewers expressed serious concerns about the paper. In particular, Reviewer #2 indicates that the model dynamics needs to be compared with dynamics of similar simple models, which involve just competition between resistant and sensitive cells, without invoking induction of stemness and resistance.

Reviewer #3 expresses very serious concern about ability of using minimal semi-mechanistic model for providing meaningful predictions without calibration/validation against an experimental dataset. 

We cannot make any decision about publication until we have seen the revised manuscript and your response to the reviewers' comments. Your revised manuscript is also likely to be sent to reviewers for further evaluation.

Sincerely,

Mark Alber, Ph.D.

Deputy Editor

PLOS Computational Biology

Mark Alber

Deputy Editor

PLOS Computational Biology

Reviewer's Responses to Questions

**Comments to the Authors:**

Reviewer #1: The authors introduce a novel mathematical model of treatment that pushes cells either into death or an increased stem-like state. The simple model is analyzed mathematically to provide approximations for time to progression (TTP), and an analysis of the important pharmacological parameters which determine when an optimal dose exists below the max tolerable dose (MTD). In general, the manuscript is interesting and well-conceived, and PLOS Computational Biology is a suitable journal for this type of manuscript.

The main result from the model is that there is non-monotonicity of eigenvalues (and therefore, there exists an optimal drug dose) only when potency to induce drug resistance is low.

Minor concerns:

1. Authors note that figure 3A & 3C are non-monotonic in lambda 1. Is this true? It’s difficult to tell visually from the figures, as it appears that lambda 1 plateaus to a lower bound. In this case, it might be helpful to pick an exaggerated example.

2. Can the authors please describe the reasoning behind displaying the summation “kSR + kRS” in figure 3. This represents the rate of flow between Resistant/Sensitive, added to the rate of flow between Sensitive/Resistant. Is this correct, and why is the addition important quantity (and not, for example, the net total rate of Sensitive: gS - kRS).

If I understand correctly, kRS is a constant (not a function of drug dose), as it does not appear in figure 2.

3. I think it would be helpful to place at least one of the supplementary figures (S1 through S4) inside the main text, to show example trajectories under treatment.

4. Figure 4 is a nice result, although I wonder if the result can be simplified. Perhaps this simulation could be simplified for plotting the dose at which max value of TTP occurs, across the range of values for PSR and ESR. If the corresponding dose is equivalent to MTD (100), then it’s implied to be monotonic.

5. The point in the discussion about adaptive therapy is intriguing. While it is certainly true that the MTD dose for cases A & C result in inferior TTP, it does not necessarily follow that adaptive therapy would outperform MTD here, because adaptive approaches often administer MTD dosing, but with periodic holidays. Perhaps it would be good to note the worthwhile treatment options of both adaptive therapy and metronomic, or other dose minimization strategies, given that there appears to be an optimal dose, m, in cases A & C.

6. Please place a box around the legend in Figures S1-4 (right side of panel A), because it makes it appear that the legend marker is a data point which doesn’t fall on the black line.

Reviewer #2: This is an interesting paper, from a prolific and highly-powered research team, which uses an extremely simple linear ODE model to depict the limits of chemotherapy, under assumption that chemotherapy leads to induction of resistance and does not just select out pre-existing resistant cells. Analysis of the model leads to the conclusion that dynamics involves a saddle point and that 4 qualitatively different regimes of treatment can be distinguished, leading to different life outcomes for the patient. If proven true, and substantiated, this may be useful as an advice to physicians and cancer counselors, and of course patients.

Presentation of the model and results is preceded by an interesting introductory review explaining the ideal of "stemness", which is a reversal of differentiation leading potentially to evolution of resistance. These are important and interesting concepts, even if many of these indicate that stemness is a feature usually present as a continuum of types (cf. the papers from Herbert Levine's group related to EMT), and not 0/1 as it is effectively assumed in the model.

In this reviewer's opinion, the paper is interesting but a thorough treatment of several important issues is missing.

1) How the dynamics of the model compares to similarly simple models, which involve just competition between resistant and sensitive cells, without invoking induction of stemness and resistance. It will be interesting and important to show that this model results in dynamics different from the almost equally simple nonlinear ODE model of lung cancer treatment by Kozlowska et al. (PLoS CB 2020), which involves only selection of pre-existing resistant cells. Will such not lead to trajectories similar as in the "saddle-point" set-up?

(2) Enough is know by now about progression dynamics of many types of cancer under treatment, to endow the model with parameters more thoroughly grounded in data. Kozlowska et al. paper on lung cancer has such estimates, and an earlier paper in Cancer Research with the same first author features estimates for ovarian cancers. The same is the case regarding the another Cancer Research paper from Bozic lab concerning colon cancer. Which of the 4 regimes will prevail under data from these papers?

(3) In the older literature, gradual evolution of resistance was considered, such as resistance to methotrexate induced by DHFR gene amplification. How will performance of the current model change under gradual evolution of ? A thorough study would require far-reaching generalization of the model, but it is possible that relevant mathematicsl results exist in the literature

It seems fair to request at least a discussion of these issues.

The paper is generally very well-written, although the term "baked-in" seems a bit idiosyncratic compared to the more common "wired-in".

Reviewer #3: In this work the authors study a very important and timely question: how treatment response and recurrence are influenced by the dual impact of cytocidal treatments: the killing impact of the drug vs the propensity for the drug to induce the transition to a resistant phenotype. By introducing a generic elementary model, the authors undertake a quantitative analysis of how the propensity of the drug to promote death versus a transition to resistance influences the often-used clinical measurement of median time to progression (TTP). The authors perform a very reasonable set of analyses on their model, centered around the question of how a drug’s capacity to induce resistance influences TTP. While I find the quantification of the results to be interesting, I am left wondering how much impact the paper will have given the simplistic nature of the model, the lack of calibration to data, and the fairly intuitive nature of many of the results. I expand upon my concerns below.

Major comments

• I always appreciate working with a minimal semi-mechanistic model. What I wonder here is: is this model too simple to provide meaningful predictions? I would be less concerned about the simplicity of the model if there was some calibration/validation against an experimental dataset. But, in the absence of this, I’m left wondering: 1) is the assumption of exponential growth sufficient, and 2) are the functional forms in eqn. (4) reasonable choices to model the impact drug dose has on the rates of cell-kill and transition-to-resistance?

• The TTP analysis is very nicely presented. I guess I’m just stuck wondering: what has been gained from the analysis? I absolutely think the message that designing a dosing strategy cannot be optimally done without understanding the propensity of the drug to induce resistance is an important one. But, if one accepts the evidence of drug-induced resistance, is it not self-evident that optimal dosing strategies would be influenced by this drug-induced phenotypic transition? If the model were grounded in data, that would be different, as then directly actionable suggestions could be made. But without that, the result really is: induced resistance can impact TTP. No doubt I agree, but I agreed with this point just from reading the intro to the paper!

• Many of the qualitative conclusions made seem fairly intuitive, given the assumptions built into the model. While more quantitative aspects of the conclusions are certainly not intuitive, those are also a lot more dependent on functional form and the somewhat arbitrary values parameters were set to. In particular, the following did seem intuitive without an analysis of the model: 1) In the case of high efficacy of induction (Cases A, B), there is diminishing TTP gains as done increases. 2) Saturation level of TTP decreases as efficacy of resistance induction increases. The one result I found less intuitive is that non-monotonicity of TTP is only observed when the potency to induce drug resistance is low. Though, I do think it's more accurate to say "low relative to the potency of tumor killing".

Minor comments

• Lines 78-79: the authors state that “However, the intrinsic limit that induced resistance places on therapy has not been systematically evaluated.” While not as systematic analysis as done herein, this was studied in a model of induced resistance in “Mathematical Approach to Differentiate Spontaneous and Induced Evolution to Drug Resistance During Cancer Treatment” by Greene and colleagues.

• Figure 1: why isn’t the drug dose m shown influencing the death of resistant cells?

• At first I was having trouble interpreting “high potency” and “low potency”. Further (and mor careful) reading made clear that high potency means that a low concentration of drug is needed to cause the halfway response between baseline induction and maximal induction. While it makes sense to call this “high potency”, I found it confusing and thought the language could be clarified earlier in the paper.

• Line 226: “We first consider a drug with a high efficacy and low potency (i.e., high EC50) for inducing resistance to cell killing”. At first the language threw me off, because there are also terms in the model that describe the efficacy and potency of cell killing. It would be clearer to me if this just said “We first consider a drug with a high efficacy and low potency (i.e., high EC50) for inducing resistance.”

• I was wondering if the authors could explain, in Figure 3 (bottom row), why all four plots seem to first hit their maximum value around the same dose of m = 40. I would have thought that the potency parameter P_SR would have influenced this time, but maybe it’s more a function of the fixed parameters in the model?

• When I downloaded the supplemental figures, they were of low image quality, even when viewing them in a small window.

**Have the authors made all data and (if applicable) computational code underlying the findings in their manuscript fully available?**

Reviewer #1: Yes

Reviewer #2: Yes

Reviewer #3: Yes

PLOS authors have the option to publish the peer review history of their article (what does this mean?). If published, this will include your full peer review and any attached files.

Reviewer #1: **Yes: **Alexander R. A. Anderson

Reviewer #2: **Yes: **Marek Kimmel

Reviewer #3: No
---

## [Decision Letter · Decision Letter 1]

20 Jun 2022

Dear Ms. Angelini,

We are pleased to inform you that your manuscript 'A model for the intrinsic limit of cancer therapy: duality of treatment-induced cell death and treatment-induced stemness' has been provisionally accepted for publication in PLOS Computational Biology.

Best regards,

Mark Alber, Ph.D.

Deputy Editor

PLOS Computational Biology

Mark Alber

Deputy Editor

PLOS Computational Biology

Reviewer's Responses to Questions

**Comments to the Authors:**

Reviewer #1: RE: #1 – thank you; the insets much improve the manuscript.

RE: #2 – thank you; the clarification added to the manuscript regarding k_RS is helpful.

All other concerns were adequately addressed. The manuscript is much improved after this round of review.

Reviewer #3: The authors' responses and edits have given me a better appreciation of the work done. I especially like the addition of the section on virtual cohort simulations. I am left with just a few minor questions:

- Line 224: the line "we anchor the pharmacodynamics for killing of sensitive and resistant cell..." is unclear to me. Am I correct that this means that the PD parameters in ds(m) and dR(m) are fixed?

- Line 293: I now understand how to read this, but my brain still freezes when I read the statement "remission is prolonged under treatment by a low efficacy drug". For me, it is clearer if it read "remission is prolonged under treatment with low efficacy for inducing resistance, as compared to treatment with a drug with a high efficacy for inducing resistance." Or, something like that!

- A typical step in the generation of a virtual population cohort, when sampling from a uniform distribution, has become to optimize the parameters so that all growth curves fall within some pre-determined number of standard deviations from the expected growth curve (see: Efficient Generation and Selection of Virtual Populations in Quantitative Systems Pharmacology Models by Allen et al, 2016). Otherwise, virtual patients generated from randomly sampling uniform distributions can have dynamics very unlikely to occur in a real patient. Though, I grant, the ranges considered for most parameters is quite small, so maybe this step wasn't needed. Curious to know the authors' thoughts.

A few typos noticed:

- Line 220: "a lowering the resistant cell death"

- Line 230 "alteringg"

- Line 460: missing period

**Have the authors made all data and (if applicable) computational code underlying the findings in their manuscript fully available?**

Reviewer #1: Yes

Reviewer #3: Yes

PLOS authors have the option to publish the peer review history of their article (what does this mean?). If published, this will include your full peer review and any attached files.

Reviewer #1: **Yes: **Alexander Anderson

Reviewer #3: No

---

## [Editor Report · Acceptance letter]

19 Jul 2022

PCOMPBIOL-D-21-02030R1 

A model for the intrinsic limit of cancer therapy: duality of treatment-induced cell death and treatment-induced stemness

Dear Dr Huang,

I am pleased to inform you that your manuscript has been formally accepted for publication in PLOS Computational Biology. Your manuscript is now with our production department and you will be notified of the publication date in due course.

With kind regards,

Zsofia Freund
